# DISTRIBUTED LINEAR DIMENSIONALITY REDUCTION ASSISTED BY CENTRALIZED NN FOR CLASSIFICATION

## ABSTRACT

Linear dimensionality reduction is a widely used technique in data compression, especially under computationally-constrained platforms. This paper presents a linear dimensionality reduction technique tailored for distributed edge devices, balancing resource constraints like data-rate and computing power at the device side, while ensuring high classification accuracy at the server side. The core concept of our approach is the simultaneous training of a unique single-layer for each distributed device, determined by its compression needs, coupled with a centralized deep neural network on the server for all-device classification. A standout feature of our approach is its adaptability: when integrating a new device aiming to compress data in an untrained dimension, only minimal training for the device's initial two layers is needed, leaving the server's centralized deep neural network and the compression layers for all existing devices untouched. Additionally, our findings indicate that the peak accuracy attainable through our method approaches that of the optimal accuracy achievable by the ideal Maximum Likelihood classifier, outperforming traditional matrix decomposition-based techniques like Principal Component Analysis (PCA) and Linear Discriminant Analysis (LDA). Compared to distance-metric-based strategies like Neighborhood Component Analysis (NCA), our technique offers a marked reduction in training complexity for large datasets. Experimental studies show that our approaches result in significant improvements in classification accuracy under the same data-rate requirements compared to existing linear dimensionality reduction approaches on real data sets.

## 1 INTRODUCTION

We are witnessing an unprecedented growth in the number of low-cost devices being deployed for data collection at the network edge fueled by an Internet of Things (IoT) revolution El-Tawab et al. (2017); Tapashetti et al. (2016); Payero et al. (2017). However, these edge devices typically have limited computational resources, which makes it challenging to locally execute complex machine learning algorithms for data processing. Consequently, it becomes essential to transmit the collected information to a server with ample computational capability for further processing. Moreover, due to both rate limitations at the low-cost devices and data-rate constraints of the wireless channel that may be handling many concurrent transmissions, it becomes crucial for these devices to compress the data prior to transmission in order to conserve bandwidth. The server can then efficiently process this compressed data. The emphasis of this paper is on this process of data compression at the distributed edge device, also known as *dimensionality reduction* at the network edge.

Among various dimensionality reduction methods, linear dimensionality reduction, which linearly projects high-dimensional data onto a lower-dimensional space, is a popular choice due to its computational efficiency and robustness against overfitting Goldberger et al. (2004). Given the constrained computational capabilities of low-cost IoT devices, linear dimensionality reduction emerges as a practical choice. Therefore, this study predominantly centers on leveraging this technique to tackle classification challenges at the network edge.

A comprehensive survey of linear dimensionality reduction techniques can be found in Cunningham & Ghahramani (2015). Among these techniques, matrix decomposition-based approaches, such as Principal Component Analysis (PCA) Pearson (1901) and Linear Discriminant Analysis (LDA) Fisher (1936), are the most well-known and widely adopted. Distance-metric based learning is also a widely

studied category of linear dimensionality reduction methods for classification. Notable examples of distance-metric based learning approaches include Neighbourhood Components Analysis (NCA) Goldberger et al. (2004) and Large Margin Nearest Neighbor (LMNN) Weinberger et al. (2005). Alternative methods for linear dimensionality reduction include Canonical Correlations Analysis (CCA) Hotelling (1992), Maximum Autocorrelation Factors (MAF) Larsen (2002), Slow Feature Analysis (SFA) Wiskott & Sejnowski (2002), Independent Component Analysis (ICA) Stone (2002), and many others.

In this paper, we introduce a novel imbalanced NN-based linear dimensionality reduction technique tailored for distributed edge devices with low computational capability. The core concept of our approach is the simultaneous training of a unique single-layer for each distributed device, determined by its compression needs, coupled with a centralized deep neural network on the server for all-device classification. This approach builds on the concept of *split computing* found in existing literature, where edge devices deploy a streamlined NN for data compression to preserve communication bandwidth Matsubara et al. (2022; 2019). On the one hand, we specialize to the case of linear operations at the edge device (due to its limited computational capabilities). On the other hand, we extend it to a *tree-structured* split computing that uses a shared server-side NN, combining input that comes from distributed devices with varying compression levels (see Fig. 1).

To get clearer and more analyzable results, we first look at the special case where there is only a single device in the network. Our results demonstrate that the peak accuracy attainable through our imbalanced NN-based method approaches that of the optimal accuracy achievable by the ideal Maximum Likelihood (ML) classifier. Furthermore, we show that no nonlinear activation function should be applied following the first fully-connected layer. We also revisit matrix decomposition-based approaches and distance metric learning-based approaches. We delve into two widely employed matrix decomposition-based techniques - Principal Component Analysis (PCA) and Linear Discriminant Analysis (LDA) - and provide a theoretical analysis of their performance, which is shown to be inferior in terms of accuracy compared to our proposed Neural Network (NN)-based method. Regarding the distance-metric based learning approaches such as Neighbourhood Components Analysis (NCA), we demonstrate that our NN-based method offers considerably lower training complexity when dealing with large datasets.

We then extend our design from the single-device scenario to accommodate a more generalized case involving multiple distributed devices. In an effort to reduce server storage requirements and increase the pool of training data without compromising data security, we advocate for the use of a singular, centralized NN for classification across all dimensions, rather employing separate NNs for each device. A standout feature of our approach is its adaptability: when integrating a new device aiming to compress data in an untrained dimension, only minimal training for the first two layers of the new device is needed, leaving the server's centralized deep neural network and the compression layers for all existing devices untouched. Notably, our strategy not only economizes on resources for training but also minimizes disruption to the workflow of previously integrated devices.

Through rigorous experimental studies, we show that our methods yield significant improvements in classification accuracy under identical data-rate constraints, compared to existing linear dimensionality reduction approaches. These enhancements are evident across real-world datasets and are applicable for both single-device and multiple-device scenarios.

## 2 SYSTEM MODEL AND PROBLEM STATEMENT

Consider a data classification framework deployed at the network edge, as illustrated in Fig. 1. There are $N$ edge devices that collect samples from the environment for classification purpose. Given the constrained computational capabilities of the low-cost chip embedded in each edge device, local classification is not feasible. Thus, these samples are transmitted to a edge compute server with a high-performance computing cluster for subsequent classification. As highlighted in the introduction, the constraints on data transmission rates mandate the compression of data before its transmission. Furthermore, due to the diverse channel conditions encountered by various devices, the compression rates differ among devices. Specifically, a device with good channel conditions might employ a less aggressive compression rate, and vice versa. It is assumed that the raw samples harvested by all devices share a common input dimension, but due to different compression rates, the compressed data to be transmitted over the channel from different devices may have different dimensions. Once

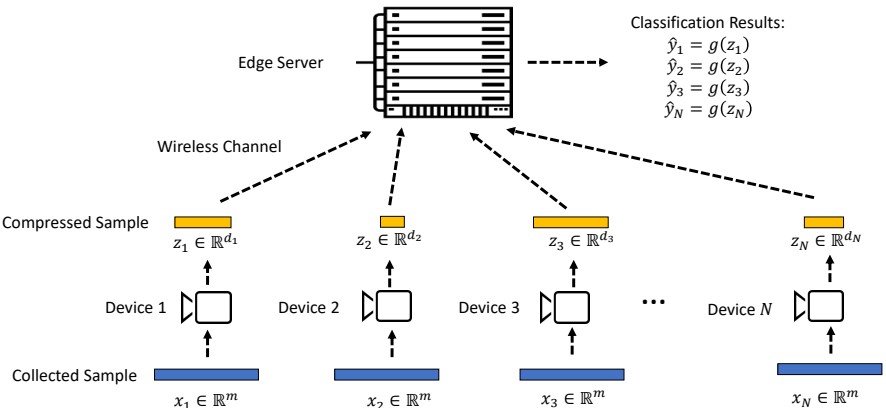

Figure 1: System model for data classification at network edge.

the edge server receives the compressed data, a classification algorithm processes and classifies them based on the compressed data received.

In this paper, we explore *linear dimensionality reduction* as the compression mechanism at the edge devices, due to its low complexity and robustness characteristics (as discussed in the introduction). Let $x_i$ denote a sample collected by source $i$ ($i = 1, 2, \cdots, N$). Assume there are $L$ classes of samples, and each $x_i$ has a corresponding label $y_i$, where $y_i \in [L] \triangleq \{1, 2, \cdots, L\}$. Let $m$ denote the dimension of the collected input sample, i.e., $x_i \in \mathbb{R}^m$ for each $i = 1, 2, \cdots, N$, and let $d_i$ denote the dimension after compression at device $i$, i.e., $z_i \in \mathbb{R}^{d_i}$. Let $\boldsymbol{W}_i \in \mathbb{R}^{d_i \times m}$ denote the linear projection matrix at device $i$. Consequently, the compressed sample derived from $x_i$, represented by $z_i$, is given by:

$$z_i = \boldsymbol{W}_i x_i, \ i = 1, 2, \cdots, N. \tag{1}$$

A classification algorithm at the server, denoted by $g(\cdot)$, uses the compressed samples for classification, i.e.,

$$\hat{y}_i = g(z_i), \ i = 1, 2, \cdots, N, \tag{2}$$

where $\hat{y}_i$ is the classified label of the sample $x_i$.

We assume that the classification tasks across all devices are uniform, meaning every sample should be categorized into one of the $L$ classes, just like the assumption in federated learning Li et al. (2020); Zhang et al. (2021). While the classification objective remains constant, the classification algorithm $g(\cdot)$ must accommodate varying input dimensions, namely, $d_1, d_2, \ldots, d_N$.

We assume that each device $i$ has a set of training samples and test samples, denoted by $\mathcal{X}_i^{\text{train}}$ and $\mathcal{X}_i^{\text{test}}$, and the corresponding sets of labels, denoted by $\mathcal{Y}_i^{\text{train}}$ and $\mathcal{Y}_i^{\text{test}}$, respectively. The classification accuracy in the training phase for device $i$ is defined as

$$\alpha_i^{\text{train}} = \frac{1}{|\mathcal{X}_i^{\text{train}}|} \sum_{(x_i, y_i) \in (\mathcal{X}_i^{\text{train}}, \mathcal{Y}_i^{\text{train}})} \mathbb{1}[g(\boldsymbol{W}_i x_i) = y_i]. \tag{3}$$

And the average accuracy in the training phase for all devices is defined as

$$\bar{\alpha}^{\text{train}} = \frac{1}{N} \sum_{i=1}^{N} \alpha_i^{\text{train}}. \tag{4}$$

In the training phase, our target is to find the optimal $\boldsymbol{W}_i$'s and $g(\cdot)$ that maximize $\bar{\alpha}^{\text{train}}$. After that, we apply these $\boldsymbol{W}_i$'s and $g(\cdot)$ to the test datasets, and the classification accuracy of the test set for device $i$ is given by

$$\alpha_i^{\text{test}} = \frac{1}{|\mathcal{X}_i^{\text{test}}|} \sum_{(x_i, y_i) \in (\mathcal{X}_i^{\text{test}}, \mathcal{Y}_i^{\text{test}})} \mathbb{1}[g(\boldsymbol{W}_i x_i) = y_i]. \tag{5}$$

And the average accuracy in the test phase for all devices is defined as

$$\bar{\alpha}^{\text{test}} = \frac{1}{N} \sum_{i=1}^{N} \alpha_i^{\text{test}}. \tag{6}$$

Our final target is to find $\boldsymbol{W}_i$'s and $g(\cdot)$ that maximize $\bar{\alpha}^{\text{test}}$.

To solve this problem, in the rest of this paper, we will begin with the special case where there is only a single device in the network. This approach will lay a foundational groundwork that shapes our broader exploration. With insights from this special case, we will then transition to the distributed devices context, ensuring that methodologies derived for a single device could be effectively extended to accommodate multiple devices.

## 3 SINGLE-DEVICE CASE: AN IMBALANCED NN-BASED SOLUTION

Initially, we examine a specific where $N = 1$, indicating the presence of just a single device in the network. For this particular scenario, in this section, we introduce a novel "Imbalanced NN-based" approach to determine the dimensionality reduction matrix $\boldsymbol{W}$ and the classification algorithm $g(\cdot)$, conduct theoretical analysis, and compare it with existing linear dimensionality reduction methods.

### 3.1 IMBALANCED NN-BASED APPROACH: BASIC FRAMEWORK

Fig. 2 presents an example framework for the imbalanced NN-based approach for a single device. The framework consists of two components:

1. An $m \times d$ fully connected layer, which represents the linear dimensionality reduction matrix $\boldsymbol{W}$.

2. A deep neural network, embodying the non-linear classification algorithm $g(\cdot)$.

Since the complexity of the first component is much lower than that of the second component, we refer to this design as "imbalanced."

**Imbalanced NN Training Operation:** During the training phase, both $\boldsymbol{W}$ and $g(\cdot)$ are trained together using the backpropagation algorithm and cross-entropy loss function, with input from $x_i^{\text{train}}$'s and $y_i^{\text{train}}$'s. Upon convergence, $\boldsymbol{W}$ is isolated as the linear dimensionality reduction matrix and will be deployed at the edge device.

It is important to note that the proposed framework does not dictate the specific design of the deep neural network $g(\cdot)$. Essentially, $g(\cdot)$ can be any neural network with a minimum of two layers and at least one non-linear activation function. In Fig. 2, we illustrate a particular design of the deep neural network $g(\cdot)$, which consists of $B$ blocks with identical structures: a series of Multilayer Perceptrons (MLPs) featuring GELU activation Hendrycks & Gimpel (2016) and skip connections He et al. (2016). This design draws inspiration from the MLP-Mixer Tolstikhin et al. (2021), which has demonstrated state-of-the-art performance on manyof real-world datasets, compared with both ResNets and Transformers. Unless otherwise stated, all numerical results in this paper are based on the design depicted in Fig. 2.

It is also crucial to note that no non-linear activation function should follow the first fully connected layer (represented by $\boldsymbol{W}$). The rationale behind this design choice will be elaborated upon in the subsequent section.

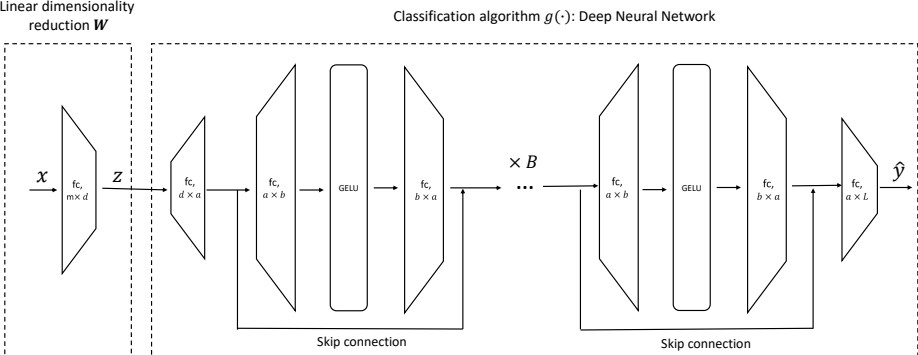

Figure 2: An Example for Imbalanced NN-Based Linear Dimsionality Reduction for a single device.

### 3.2 GLOBAL OPTIMALITY

In this section, we analyze the global optimality of the imbalanced NN-based approach. We assume the availability of infinite training and testing data. Under this assumption, both the training and testing sets can be characterized by a Probability Density Function (PDF). Moreover, we assume that the PDFs of the training and testing sets are identical. Consequently, the accuracy of the training and testing sets are equal. The probability of label $y$ given a data sample $x$ is denoted as $p(y|x)$, and the PDF of class $i$ is denoted by $f(x|y = i)$.

We now investigate the global optimal accuracy over all feasible approaches (including a linear dimensionality reduction to $\mathbb{R}^d$ and a general classification algorithm) without being limited to the NN-based method. For any fixed $\boldsymbol{W}$, the ideal (optimal) classification algorithm $g(\cdot)$ corresponds to the Maximum Likelihood (ML) algorithm, expressed as:

$$g_{\text{ML}}(x) = \arg \max_y p(y|\boldsymbol{W}x). \tag{7}$$

The accuracy of this ML classifier can be represented as:

$$\alpha_{\text{ML}}(\boldsymbol{W}) = \int f(x) \cdot \max_y p(y|\boldsymbol{W}x) \cdot dx. \tag{8}$$

Consequently, the global optimal accuracy over all feasible approaches under a fixed $d$ is given by:

$$\alpha^*(d) = \max_{\boldsymbol{W}} \alpha_{\text{ML}}(\boldsymbol{W}), \ \boldsymbol{W} \in \mathbb{R}^{d \times m}. \tag{9}$$

Returning to our NN-based approach, where the classification algorithm $g(\cdot)$ is a deep neural network, let us denote $\alpha^*(d, g^{\text{NN}})$ as the global optimal accuracy when $g(\cdot)$ is constrained to a specific neural network structure $g^{\text{NN}}$ (e.g., the neural network illustrated in Fig. 2). The following theorem establishes the relationship between $\alpha^*(d, g^{\text{NN}})$ and $\alpha^*(d)$:

**Theorem 1** *If $f(x|y = i)$ is continuous with respect to $x$ for each $i = 1, 2, \cdots, L$, then for any $\epsilon > 0$, there always exists a neural network structure $g^{NN}$ such that $\alpha^*(d, g^{NN}) > \alpha^*(d) - \epsilon$.*

Theorem 1 indicates that the global optimal accuracy of the NN-based approach can be arbitrarily close to the global optimal accuracy over all approaches. The underlying concept of this theorem stems from the universal approximation ability of neural networks Cybenko (1989); Hornik et al. (1989), which enables them to approximate the ML classifier when $f(x|y = i)$ is continuous with respect to $x$. A detailed proof of Theorem 1 is provided in the Appendix.

Now we discuss why a non-linear activation function should not follow the first fully connected layer. Let $\alpha^*(d, \sigma, g^{\text{NN}})$ denote the global optimal accuracy when a non-linear activation function $\sigma(\cdot)$ is applied after the first fully connected layer and is followed by a neural network $g^{\text{NN}}$. The following proposition states the relationship between $\alpha^*(d, \sigma, g^{\text{NN}})$ and $\alpha^*(d, g^{\text{NN}})$.

**Proposition 1** *For any activation function $\sigma$, we have $\max_{g^{NN}} \alpha^*(d, \sigma, g^{NN}) \leq \max_{g^{NN}} \alpha^*(d, g^{NN})$.*

The rationale behind Proposition 1 is that the neural network $g^{\text{NN}}$ can approximate any continuous function, including the activation function $\sigma$. Therefore, the activation function $\sigma$ cannot enhance the global optimal accuracy. On the other hand, activation functions applied to the reduced-dimension data $z$ might be unrecoverable by the neural network $g^{\text{NN}}$, potentially degrading performance. For instance, with the ReLU activation, the negative part of the signal is irretrievable. A proof of Proposition 1 is provided in the Appendix. Numerical results in Appendix D will validate the intuition that some non-linear activation functions may worsen the performance.

It is important to note that although the global optimal accuracy achieved by our imbalanced NN-based method can be arbitrarily close to the global optimal accuracy attainable by all other feasible approaches, this does not guarantee that the backpropagation algorithm will converge to the global optimum, as the cost function is generally non-convex so the algorithm may converge to local minimums or saddle points, as is predominantly the case for other classification algorithms.

### 3.3 Comparison with Matrix Decomposition Based Approaches

Matrix decomposition-based methods are traditionally favored for linear dimensionality reduction due to their computational efficiency and straightforward geometric interpretations. The two most widely adopted matrix decomposition-based approaches are Principal Component Analysis (PCA) Pearson (1901) and Linear Discriminant Analysis (LDA) Fisher (1936).

Despite the widespread use of PCA and LDA for linear dimensionality reduction, their application in our edge classification problem often results in significantly lower classification accuracy compared to our imbalanced NN approach. This claim will be substantiated by numerical results on various real-world datasets in Section 5.

From the theoretical perspective, let us revisit Theorem 1. The global optimal accuracy of the NN-based approach can approach $\alpha^*(d)$ (the global optimal accuracy over all methods), as the NN classifier $g^{\text{NN}}$ has the capacity to approximate the ideal (Maximum Likelihood) classifier. However, for PCA or LDA, even if the classifier $g(\cdot)$ is ideal, the accuracy may still fall short of $\alpha^*(d)$. This is due to the fact that the linear projection of PCA or LDA may degrade the accuracy compared to the optimal projection. Illustrative examples are given in Appendix.

### 3.4 Comparison with Distance-Metric Based Learning Approaches

Distance-metric based learning is also a widely studied category of linear dimensionality reduction methods for classification. These methods aim to optimize the matrix $\boldsymbol{W}$ in order to improve certain objectives within the training data, based on the Mahalanobis distance $d_{\boldsymbol{W}}(x_i^{\text{train}}, x_j^{\text{train}}) = ||\boldsymbol{W}x_i^{\text{train}} - \boldsymbol{W}x_j^{\text{train}}||_2^2$. They typically utilize gradient-descent based optimization algorithms for training. Notable examples of distance-metric based learning approaches include Neighbourhood Components Analysis (NCA) Goldberger et al. (2004), Large Margin Nearest Neighbor (LMNN) Weinberger et al. (2005), and others. NCA optimizes the leave-one-out KNN score on the training set, and it often outperforms PCA and LDA in terms of classification accuracy on many real-world datasets.

However, distance-metric based learning approaches have a significant drawback - their training complexity is considerably high, particularly with large datasets. This is due to their objective function incorporating the distance between any two samples in the training set, leading to a total of $\frac{N_{\text{train}}^2 - N_{\text{train}}}{2}$ distance metrics and rendering the complexity for each training iteration at least proportional to $N_{\text{train}}^2$. Even faster variants, such as Fast Neighbourhood Components Analysis (FNCA) Yang et al. (2012), still require computing of all the distance metrics, thereby maintaining a complexity proportional to $N_{\text{train}}^2$. Although training is executed on the server side, the process can be overly time-consuming when dealing with large datasets.

In contrast, our proposed imbalanced NN approach uses backpropagation to train the neural network, resulting in a complexity per iteration that is linear with $N_{\text{train}}$. Thus, it is evident that for large datasets, the imbalanced NN approach's training complexity is significantly lower than that of distance-metric based learning approaches.

When it comes to accuracy performance, as demonstrated in Section 5, our imbalanced NN approach displays a slightly superior accuracy to distance-metric based learning approaches on real data sets.

## 4 Multi-Device Case: A Distributed Approach

Having established the imbalanced NN-based approach for a single device, we now aim to extend this method to accommodate an edge network comprising $N$ devices, as in the scenario depicted in Fig. 1. Each of these devices has its distinct compression dimension $d_i$.

The basic approach for this extension would be to train $N$ distinct NNs, a different one for each edge device that is tailored for its particular compression dimension $d_i$. While this strategy is feasible, it is accompanied by a couple of challenges:

1. Storing $N$ separate, large-scale NNs on the server would consume substantial storage, proving to be resource-intensive.

2. In this strategy, training processes for the NN of each device are isolated. If devices choose not to share their training data with one another due to privacy implications, it could compromise the training efficacy, possibly leading to insufficient training data.

To address these challenges, we advocate for the utilization of a common deep neural network to serve as the classifier for all devices, as illustrated in Fig.3. It is crucial to understand that this single NN is designed to manage multiple input dimensions, namely $d_1, d_2, \cdots, d_N$. Consequently, the first layer of this single NN acts as a transition layer, converting these diverse input dimensions to a uniform dimension $a$, as shown in Fig.3. Additionally, as indicated by Proposition 1, there is no requirement for any activation function preceding the transition layer.

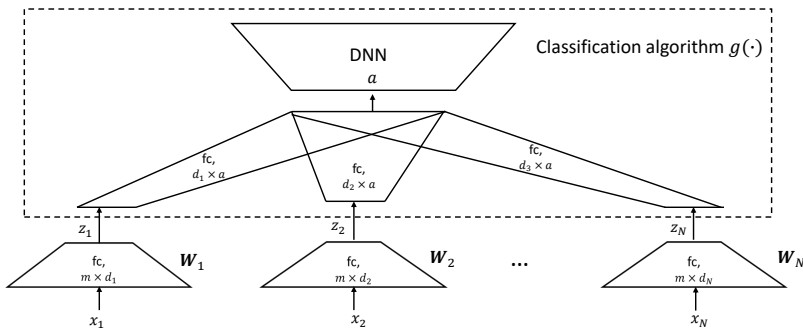

Figure 3: An Example for Imbalanced NN for multiple devices.

**Training for Multiple Devices:** Just like the single-device case, in the training phase for multiple distributed devices, the classification algorithm $g(\cdot)$ and the linear compression matrices $\boldsymbol{W}_i$'s are trained together. In each training iteration, every device updates its compressed samples based on its respective compression matrix, $\boldsymbol{W}_i$. Subsequently, the server deploys backpropagation to calculate the gradients for both $g(\cdot)$ and $\boldsymbol{W}_i$ matrices. It then returns the gradients of $\boldsymbol{W}_i$ for each device $i$ via a downlink channel, facilitating the update of its local weights. For the purpose of optimizing aggregate accuracy, the server requires data from all devices in every iteration (or batch) to ascertain the gradients. Throughout this procedure, samples from all devices jointly aid in updating the weights for the classification algorithm $g(\cdot)$, ensuring that no information is inadvertently shared among the devices.

In contrast to training $N$ separate NNs for each device, the above procedure offers dual benefits: (i) it decreases storage demands on the server, and (ii) it augments the volume of training data without risking data leakage. As showcased in Section 5, this method yields similar accuracy performance comparable to that of training $N$ individual NNs for every device.

**Integrating New Device with Untrained Dimension:** In practical network environments, it is typical for new devices to join or for existing ones to depart. Ensuring a seamless integration for these new devices is paramount from a systems perspective.

If the incoming device possesses a compression dimension identical to one already in the network, the integration process becomes relatively straightforward. The new device can simply request and duplicate the compression matrix from its existing counterpart. Challenges arise, however, when the new device seeks to compress its sample to a dimension that has not been trained for. One straightforward but inefficient solution is to re-train the entire network for all devices, consequently deriving new $\boldsymbol{W}_i$'s and $g(\cdot)$. This method, aside from being resource-intensive, also disrupts the operational flow of established devices.

Fortunately, a more streamlined solution is at our disposal to tackle this challenge. Let's assume we've already established an imbalanced NN framework accommodating multiple dimensions $d_1, d_2, \cdots, d_N$. When a new device with an untrained dimension $d_{N+1}$ comes into play, we can maintain the $g(\cdot)$ and the $\boldsymbol{W}_i$'s of all pre-existing devices as they are. The adjustment would involve solely re-training two layers ($m \times d_{N+1}$ and $d_{N+1} \times a$) specific to the new device. The feasibility of this approach hinges on the fact that the current $g(\cdot)$ has effectively assimilated generalized classification information across various dimensions and can aptly extend this knowledge to a fresh dimension. As

evidenced by the numerical findings in Section 5, training just these two layers for the new device achieves an accuracy level that is close to the outcome of training the entire NN with the new device.

## 5 NUMERICAL RESULTS

In this section, we assess the performance of our Imbalanced NN-based approach on real-world datasets, spanning both single-device and multiple-device scenarios. Due to space limitation, we show the results for the Extended Yale Face Database B Belhumeur et al. (1997) in the main text, and leave the results for the MNIST handwritten digits LeCun (1998) in the Appendix.

### 5.1 EXPERIMENT SETUP

All experiments in this section were conducted using an NVIDIA RTX 3090 GPU and an Intel Core i7-12700 CPU. The programming was done in Python 3.9 and Pytorch 2.0.

We do numerical study on The Extended Yale face data set, which contains 2414 frontal images of 38 subjects Belhumeur et al. (1997), i.e., about 64 images for each person. All the face images are cropped and resized to 32x32 pixels, following Cai et al. (2007). We divide the dataset using a 75/25 split for training and testing, respectively. For this dataset, the parameters for the imbalanced NN approach shown in Fig. 2 are set as follows: $m = 1024$, $a = 500$, $b = 2000$, $B = 8$, and $L = 38$. To train the neural networks, we utilize the SGD optimizer with a learning rate of 0.001, a batch size of 32, a weight decay of 0.0001, and a momentum of 0.9. The network is trained over 100 epochs.

At the device side, we simulate four linear dimensionality reduction algorithms, which include the imbalanced NN-based approach, PCA, LDA, and NCA. The implementation of the imbalanced NN-based linear dimensionality reduction adheres to the design depicted in Fig. 2. We implement NCA based on Scikit-learn Scikit-learn using the default parameters.

Note that our imbalanced NN approach can be used specifically to generate the linear projection matrix $W_i$'s, and subsequently the obtained projection can be employed to train alternative classification algorithms aside from Deep Neural Network (DNN). Therefore, at the server side, we simulate three classification algorithms (represented by $g(\cdot)$ in Fig. 1). These include DNN, Random Forest (RF), and K-Nearest Neighbors (KNN). For DNN, we use the design illustrated in Fig. 2. For RF, we use a forest comprised of 100 trees. For KNN, we use a configuration of $k = 10$ neighbors.

| Dimensionality | Classification Algorithm | | |
|---|---|---|---|
| Reduction | DNN | RF | KNN |
| When $d = 20$ | | | |
| Imbalanced-NN | (5.86, 1.54) | (8.18, 1.34) | (8.21, 0.76) |
| NCA | (19.00, 1.19) | (16.39, 0.54) | (12.25, 0.00) |
| PCA | (13.31, 0.80) | (22.81, 0.74) | (61.42, 0.00) |
| LDA | (19.44, 1.31) | (17.02, 0.67) | (12.25, 0.00) |
| When $d = 8$ | | | |
| Imbalanced-NN | (8.18, 0.36) | (10.50, 0.48) | (9.87, 0.97) |
| NCA | (29.44, 0.96) | (27.62, 0.80) | (23.51, 0.00) |
| PCA | (38.27, 1.31) | (57.98, 0.45) | (84.77, 0.00) |
| LDA | (38.60, 0.84) | (33.81, 0.86) | (29.97, 0.00) |

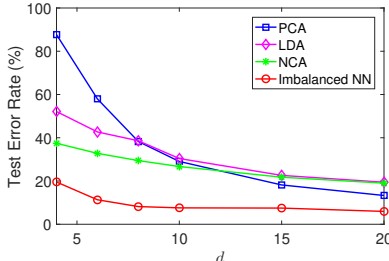

Table 1: Test error rate on Extended Yale Face Database B for the single-device case. Data format: (mean (%), standard deviation).

Figure 4: Test error rate on Extended Yale Face Database B with DNN classification algorithm for the single-device case. Every curve is the median of 5 independent runs.

### 5.2 RESULTS FOR SINGLE-DEVICE CASE

We first evaluate the performance of the imbalanced NN under the single-device special case. Table 1 shows the test classification error rate on Extended Yale Face Database B when $d = 20$ and $d = 8$ under different dimensionality reduction algorithms (imbalanced NN, NCA, PCA, and LDA) and different classification algorithms (DNN, RF, and KNN). The mean and standard deviation values are derived from five independent runs for each simulation. We can observe that, for both $d = 20$ and $d = 8$, regardless of the classification algorithm used, the average error rate when employing our

proposed imbalanced NN approach is significantly lower than those associated with NCA, PCA, and LDA. Particularly striking is the observation that, even under KNN classification–NCA's primary domain–our imbalanced NN approach outperforms NCA. Fig. 4 shows the test classification error rate on Extended Yale Face Database B for $d \in \{4, 6, 8, 10, 15, 20\}$ with DNN classification algorithm. We can observe that the imbalanced NN outperforms NCA, PCA, and LDA in terms of test error rate.

## 5.3 RESULTS FOR MULTIPLE-DEVICE CASE

In the multi-device scenario, we study a network consisting of $N = 3$ devices with distinct input dimensions: $d_1 = 4$, $d_2 = 6$, and $d_3 = 10$. A common deep neural network serves as the classifier, accommodating these diverse dimensions, as depicted in Fig. 3. For comparative analysis, we also construct three individual imbalanced NNs, each optimized for its unique dimension $d_i$. Additionally, we benchmark against other linear dimensionality reduction methods (NCA, PCA, and LDA) under the same DNN classifier. We conduct five independent runs for each simulation and calculate the mean and standard deviation values. As presented in Table 2, the test classification error rates for the three devices demonstrate that a common DNN classifier yields performance comparable to three separate NNs, and it surpasses other conventional linear techniques. Considering the storage efficiency and augmented training data without sacrificing data privacy, using a common NN emerges as the recommended choice.

| Dimensionality Reduction | Device 1 $d_1 = 4$ | Device 2 $d_2 = 6$ | Device 3 $d_3 = 10$ |
|---|---|---|---|
| Imbalanced-NN (1 common NN) | (19.43, 2.48) | (11.26, 1.75) | (7.55, 0.76) |
| Imbalanced-NN (3 distinct NNs) | (19.54, 2.66) | (11.32, 2.52) | (7.58, 2.03) |
| NCA | (37.45, 0.74) | (32.78, 0.39) | (26.69, 0.34) |
| PCA | (87.72, 0.62) | (57.95, 0.67) | (29.04, 1.83) |
| LDA | (52.12, 0.56) | (42.68, 0.54) | (30.40, 0.82) |

| Dimensionality Reduction | Device 4 $d_4 = 5$ | Device 5 $d_5 = 8$ | Device 6 $d_3 = 15$ |
|---|---|---|---|
| Imbalanced-NN (re-train 2 layers) | (13.35, 1.05) | (9.31, 0.71) | (7.85, 0.60) |
| Imbalanced-NN (re-train entire NN) | (13.04, 0.96) | (8.44, 0.85) | (7.25, 0.49) |
| NCA | (33.44, 0.60) | (29.56, 0.98) | (21.76, 0.80) |
| PCA | (75.89, 1.52) | (38.33, 1.49) | (18.18, 0.73) |
| LDA | (45.69, 0.87) | (38.60, 0.84) | (22.61, 1.25) |

Table 2: Test error rate on Extended Yale Face Database B for the multiple-device case. Data format: (mean (%), standard deviation).

Table 3: Test error rate on Extended Yale Face Database B when integrating a new device into an existing network. Data format: (mean (%), standard deviation).

Then we evaluate our algorithm's capability when integrating a new device with a previously untrained dimension. As per the prior experiment (refer to Table 2), we train a common NN that accommodates three devices with dimensions $d_1 = 4$, $d_2 = 6$ and $d_3 = 10$. Subsequently, we introduce three new devices with dimensions $d_4 = 3$, $d_5 = 8$ and $d_6 = 15$—all of which represent untrained dimensions. To integrate these devices, we apply the strategy detailed in Section 4: specifically, re-training only the initial two layers of the NN for each new device while leaving the $g(\cdot)$ and the $\boldsymbol{W}_i$'s of the existing devices untouched. For comparison, we also entirely re-train the entire NN to accommodate all devices, both old and new, and further juxtapose this against traditional linear methods (NCA, PCA, and LDA) with a DNN classifier. The results, presented in Table 2, reveal that simply re-training the initial two layers for each new device offers comparable performance to a full re-training, and outperforms standard linear approaches. Given its resource efficiency and the minimized disruption to established devices, this partial re-training strategy stands out as the optimal choice.

## 6 CONCLUSION

This paper introduces a linear dimensionality reduction method specifically designed for distributed edge devices. It addresses challenges like data-rate limitations and computational constraints on the device side while optimizing classification accuracy on the server end. In the context of a single-device scenario, our proposed imbalanced NN-based approach outperforms conventional linear dimensionality reduction techniques, both analytically and numerically. When extended to multiple devices, our methodology employs a unified, centralized NN for classification across varied dimensions, offering advantages in reduced server storage and expanded training data access without sacrificing data security. Furthermore, our framework can seamlessly incorporate new devices with previously untrained dimensions, requiring only minimal retraining and without affecting the operations of existing devices.

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

# A  APPENDIX

## A.1  PROOF OF THEOREM 1

We assume that $f(x|y = i)$ is continuous with respect to $x$ for each $i = 1, 2, \cdots, L$, and $\epsilon > 0$ is given. We consider a neural network structure $g^{\text{NN}}$ with $L$ output neurons $g_1^{\text{NN}}, g_2^{\text{NN}} \cdots, g_L^{\text{NN}}$, and let $g_i^{\text{NN}}(x)$ denote the output value of the $i$-th neuron when the input is $x$. Let $\boldsymbol{W}^*$ denote the global optimal projection matrix of the Maximum Likelihood classifier, i.e.,

$$\boldsymbol{W}^* = \arg\max_{\boldsymbol{W}} \alpha_{\text{ML}}(\boldsymbol{W}), \ \boldsymbol{W} \in \mathbb{R}^{d \times m}. \tag{10}$$

For each $i = 1, 2, \cdots, L$, since $f(x|y = i)$ is continuous with respect to $x$, the PDF of $\boldsymbol{W}^*x$, denoted by $f(\boldsymbol{W}^*x|y = i)$, is also continuous with respect to $x$. From Bayes' Theorem, we have

$$f(\boldsymbol{W}^*x) \cdot p(y = i|\boldsymbol{W}^*x) = p(y = i) \cdot f(\boldsymbol{W}^*x|y = i). \tag{11}$$

When $f(\boldsymbol{W}^*x) = 0$, the value of $p(y = i|\boldsymbol{W}^*x)$ becomes undefined, but it does not influence the classification outcome. When $f(\boldsymbol{W}^*x) > 0$, based on equation 11, it follows that $p(y = i|\boldsymbol{W}^*x)$ is continuous with respect to $x$. Then, based on the Universal Approximation Theorem, there exists a neural network $g_i^{\text{NN},0}(x)$, such that for any $x$ with $f(\boldsymbol{W}^*x) > 0$, we have

$$|g_i^{\text{NN},0}(x) - p(y = i|\boldsymbol{W}^*x)| < \epsilon. \tag{12}$$

That is, there exists a neural network $g_i^{\text{NN},0}(x)$ that can approximate the continuous function $p(y = i|\boldsymbol{W}^*x)$ with an error upper-bounded by $\epsilon$.

For the imbalanced NN classifier, the classification result is

$$g^{\text{NN},0}(x) = \arg\max_y g_i^{\text{NN},0}(x). \tag{13}$$

Let $\mathcal{X}$ denote the domain of $x$. Then the overall classification accuracy of the imbalanced NN classifier is given by

$$\alpha(d, g^{\text{NN},0}) = \int_{x \in \mathcal{X}} f(x) \max_y g_i^{\text{NN},0}(x) dx. \tag{14}$$

Plugging 12 into equation 14 and considering equation 8, we have

$$\alpha(d, g^{\text{NN},0}) > \int_{x \in \mathcal{X}} f(x) \max_y (p(y = i|\boldsymbol{W}^*x) - \epsilon) dx = \alpha^*(d) - \epsilon. \tag{15}$$

Since $\alpha^*(d, g^{\text{NN}}) \geq \alpha(d, g^{\text{NN},0})$, we have $\alpha^*(d, g^{\text{NN}}) > \alpha^*(d) - \epsilon$. This completes our proof. ∎

## A.2  PROOF OF PROPOSITION 1

Assume $\max_{g^{\text{NN}}} \alpha^*(d, \sigma, g^{\text{NN}})$ is achieved by an activation function $\sigma^*$ applied after the the first fully-connected layer and followed by a $K$-layer neural network $g_K^{\text{NN}}$, i.e., $\max_{g^{\text{NN}}} \alpha^*(d, \sigma, g^{\text{NN}}) = \alpha^*(d, \sigma^*, g_K^{\text{NN}})$. Then we consider a $(K + 1)$-layer neural network $g_{K+1}^{\text{NN}}$, where the first layer is an an identity mapping followed by an activation function $\sigma^*$, and the remaining $K$ layers are identical to $g_K^{\text{NN}}$. Let $\alpha^*(d, g_{K+1}^{\text{NN}})$ denote the the accuracy of $g_{K+1}^{\text{NN}}$ when there is no after the the first fully-connected layer in our Imbalanced NN approach. Then it is clear that $\alpha^*(d, g_{K+1}^{\text{NN}}) = \alpha^*(d, \sigma^*, g_K^{\text{NN}})$. Considering $\alpha^*(d, g_{K+1}^{\text{NN}}) \leq \max_{g^{\text{NN}}} \alpha^*(d, g^{\text{NN}})$, we have $\max_{g^{\text{NN}}} \alpha^*(d, \sigma, g^{\text{NN}}) \leq \max_{g^{\text{NN}}} \alpha^*(d, g^{\text{NN}})$. This completes our proof. ∎

## A.3  EXAMPLES FOR SUBOPTIMALITY OF PCA AND LDA

Fig. 5 presents two examples where PCA or LDA yield suboptimal results. In each figure, we generate 2-dimensional data uniformly distributed within each rectangle, and the labels of the data are indicated by different colors. In Fig. 5a, PCA projects the data onto the y-axis, which is the optimal projection. However, LDA disapproves of this projection because it causes the two purple blocks to be too distant

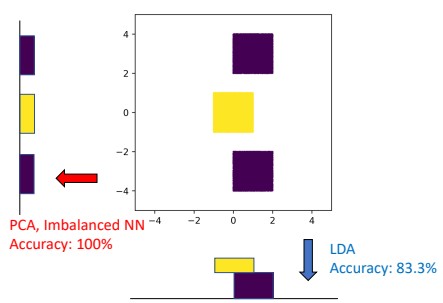 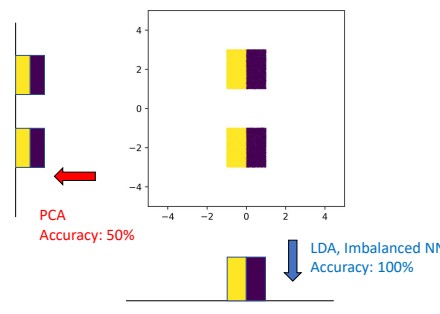

(a) Example 1: LDA is suboptimal.    (b) Example 2: PCA is suboptimal.

Figure 5: Examples to show the suboptimality of PCA and LDA.

from each other, leading to a large within-class variability. Although this within-class variability does not affect the classification accuracy, LDA rejects it, preferring instead to project the data onto the x-axis, thereby bringing the purple blocks closer together. However, this projection causes an overlap between the yellow block and the purple blocks, reducing the classification accuracy from 100% to 83.3%. On the other hand, Fig. 5b illustrates an example where LDA is optimal, while PCA falls short. The reason for PCA's suboptimal performance is straightforward: it entirely disregards the label information. As comparison, our proposed imbalanced NN-based approach, after fine-tuning, can achieve 100% accuracy in both examples, which surpasses the performance of both PCA and LDA.

### A.4 NUMERICAL RESULTS FOR MNIST DATASET

In this section, we evaluate the performance of the imbalanced NN approach for the single-device case on the MNIST dataset of handwritten digits LeCun (1998) to validate the imbalanced NN can outperform existing linear methods. The dataset consists of 60,000 training images and 10,000 test images, each image being $28 \times 28$ pixels in size. Every image in MNIST is assigned a label $y_i \in 0, 1, \cdots, 9$. For this dataset, the parameters for the imbalanced NN approach shown in Fig. 2 are set as follows: $m = 784$, $a = 500$, $b = 1000$, $B = 9$, and $L = 10$. To train the neural networks, we utilize the Stochastic Gradient Descent (SGD) optimizer with a learning rate of 0.01, a batch size of 128, a weight decay of 0.0001, and a momentum of 0.9. The network is trained over 50 epochs.

Table 4 shows the test classification error rate on MNIST when $d = 9$ under different dimensionality reduction algorithms (imbalanced NN, PCA, and LDA) and different classification algorithms (DNN, RF, and KNN). The mean and standard deviation values are derived from five independent runs for each simulation. We can observe that, regardless of the classification algorithm used, the average error rate when employing our proposed imbalanced NN approach is significantly lower than those associated with PCA and LDA.

Fig. 6 shows the test classification error rate on MNIST for $d \in \{3, 5, 7, 9, 15, 20\}$, with DNN classification algorithm. Note that LDA requires $d < L = 10$, so it cannot be applied when $d = 15$ or $d = 20$. We can observe that the error rate decreases as $d$ increases for each curve. Moreover, given the same value of $d$, the imbalanced NN outperforms PCA and LDA in terms of the test error rate.

| Dimensionality | Classification Algorithm | | |
|---|---|---|---|
| Reduction | DNN | RF | KNN |
| Imbalanced-NN | (3.92, 0.14) | (4.90, 0.12) | (3.81, 0.46) |
| PCA | (7.15, 0.23) | (9.86, 0.12) | (8.35, 0.00) |
| LDA | (8.05, 0.22) | (8.57, 0.11) | (8.00, 0.00) |

| Activation | Classification Algorithm | | |
|---|---|---|---|
| Function | DNN | RF | KNN |
| No activation | (3.92, 0.14) | (4.90, 0.12) | (3.81, 0.46) |
| GELU | (4.68, 0.18) | (5.35, 0.22) | (4.47, 0.19) |
| ReLU | (4.56, 0.22) | (5.27, 0.12) | (4.37, 0.17) |
| Sigmoid | (4.74, 0.31) | (4.60, 0.19) | (4.24, 0.19) |

Table 4: Test error rate on MNIST when $d = 9$ under different dimensionality reduction algorithms. Data format: (mean (%), standard deviation).

Table 5: Test error rate on MNIST when $d = 9$ for Imbalanced NN with different activation functions. Data format: (mean (%), standard deviation).

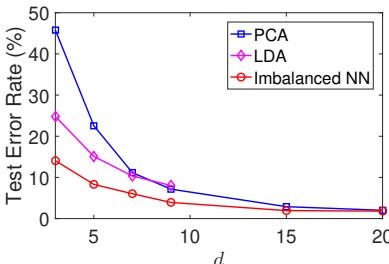 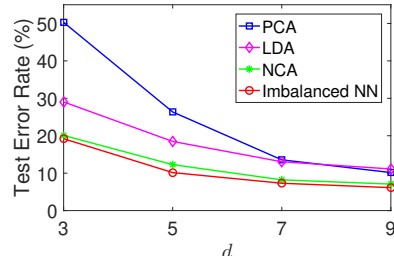

Figure 6: Test error rate on MNIST under different $d$'s with DNN classification algorithm. Every curve is the median of 5 independent runs.

Figure 7: Test error rate on Small-MNIST under different $d$'s with KNN classification algorithm. Every curve is the median of 5 independent runs.

Table 4 presents the test error rate on the MNIST dataset for the imbalanced NN approach using different activation functions: no activation, GELU, ReLU, and Sigmoid, with a dimensionality of $d = 9$. The lowest mean error rate is achieved when no activation function is applied, validating Proposition 1.

Our attempt to train NCA on the complete MNIST dataset (containing 60,000 images) resulted in a memory overflow due to its substantial space complexity. Therefore, to evaluate NCA's performance, we select a subset of 10,000 images from the MNIST dataset as the training set (termed as Small-MNIST) and use the original test set (comprising 10,000 images).

Fig. 7 illustrates the test classification error rate on Small-MNIST under different dimensions $d \in 3, 5, 7, 9$, employing the KNN classification algorithm. Although NCA outperforms both PCA and LDA, our proposed imbalanced NN approach shows a slight edge over NCA. Considering that NCA was originally developed for KNN classification, it is notable that the imbalanced NN approach slightly outperforms NCA in its area of specialization.

