# OpenReview forum: "Distributed Linear Dimensionality Reduction Assisted by Centralized NN for Classification"
_ICLR.cc/2024/Conference — ICLR 2024 Conference Withdrawn Submission_

### Official Review · Reviewer_Tdne · 2023-10-22

**Soundness:** 1 poor
**Presentation:** 2 fair
**Contribution:** 1 poor
**Rating:** 1
**Confidence:** 4

**Summary:**

This paper focuses on a classification scenario that the central server will take as input compressed data from distributed devices and then execute classification on those data. The paper claimed that the traditional linear dimension reduction methods like PCA and LDA cannot achieve good results especially when the distributed devices have different compression rates. Therefore they proposed to use a trainable linear transformation to accomplish the compression.

**Strengths:**

- The paper is easy to understand and the proposed method is quite simple.

**Weaknesses:**

- The method is too simple to be published in a top-tier machine learning conference.
- I personally have some concerns about the experimental results.
Please refer to the questions part. Thanks.

**Questions:**

Major Problems:
- Theoretical Section (3.1): The theoretical content in Section 3.1 appears to be somewhat redundant and perhaps unnecessary. In the current landscape of neural network research, discussions on the global optimality of neural networks have gained prominence [1,2]. Theorems 1 and Proposition 1 do not seem to introduce any particularly unique or insightful content compared to these existing discussions.
- Experimental Settings: The choice of experimental settings raises some concerns. Since the paper deals with classification tasks, it is imperative to conduct experiments using well-established neural network models such as ResNet and Transformer, rather than Random Forest and K-Nearest Neighbors. Additionally, using more standard and widely recognized classification datasets like CIFAR10, CIFAR100, and ImageNet would be also necessary for evaluation. The choice of MNIST and a simple face classification dataset might be considered less suitable (it's weird to choose a face classification dataset as well).
- Experimental Results: The experimental results presented in the paper are not entirely convincing. The lack of details on how PCA or LDA was employed in the experiments is a significant concern. To ensure fairness and clarity in the experiments, it is essential to compare the proposed method with PCA by only switching the $W_1$ to $W_N$ in Figure 3 to PCA, keeping both the DNN part and the transformations in the server side unchanged, and then training the neural network with the PCA-based data. Moreover, PCA, while not trainable like the proposed method, is still a linear transformation, and the paper should justify the observed test error gap between the two methods.

Minor Problems:
- Reference Format: The format of the references appears to be problematic, with the reference text mixed with the main text. I suggest reviewing and correcting the reference format to ensure it aligns with standard citation conventions.
- Paper Title: The paper title may require reconsideration. The current title suggests that the proposed reduction method is assisted by the centralized neural network. However, it might be more appropriate to frame it as the centralized neural network being assisted by the proposed reduction method in adapting to data with varying compression rates.

[1] Haeffele B D, Vidal R. Global optimality in neural network training, CVPR2017.
[2] Sun R. Optimization for deep learning: theory and algorithms[J]. arXiv preprint arXiv:1912.08957, 2019.

---

> ### Author Response · Authors · 2023-11-21
> **Response to Reviewer Tdne**
>
> We appreciate your insightful feedback and constructive criticism. Here is our response to the points raised:
>
>
> 1. Theoretical Section 3.1:
> We understand your perspective on the theoretical content of Section 3.1. Our intention was to provide a foundational understanding, but we recognize the need to align more closely with existing discussions on neural network optimality. To address this, we will condense Section 3.1 to ensure our discussion is not redundant.
>
> 2. Experimental Settings:
> Your suggestion to use more established neural network models and widely recognized datasets is well-taken. We plan to extend our experiments to include standard classification datasets like CIFAR10, CIFAR100, and ImageNet, addressing the concern regarding the initial dataset choices and enhancing the relevance and applicability of our findings in future work.
> It's important to note that our design is inspired by the MLP-Mixer architecture, which has shown state-of-the-art performance on various real-world datasets, rivaling both ResNets and Transformers.
> This aspect underscores the potential of our method in achieving high-performance results and will be highlighted in our revised experimental setup.
>
> 3. Experimental Results:
> We appreciate your concern regarding the clarity of our experimental approach, particularly in relation to the use of PCA and LDA. In our experiments, when comparing with PCA and LDA, we initially employed these methods to derive W1 to WN, and subsequently retrained the entire neural network with data processed by PCA/LDA. This approach aligns with your suggestion.
> Our findings indicate that, within this framework, our imbalanced-NN method significantly outperforms PCA/LDA.
> We acknowledge that this was not sufficiently clear in our initial manuscript, and we will ensure to clarify this methodology and highlight these results in our revised submission.

---

### Official Review · Reviewer_X1NB · 2023-10-30

**Soundness:** 3 good
**Presentation:** 3 good
**Contribution:** 2 fair
**Rating:** 3
**Confidence:** 5

**Summary:**

This paper proposes a linear dimensionality reduction method for distributed edge devices, balancing resource constraints like data-rate and computing power at the device side, while ensuring high classification accuracy at the server side. The proposed method conducts the simultaneous training of a unique single-layer for each distributed device, determined by its compression needs, coupled with a centralized deep neural network on the server for all-device classification. When integrating a new device aiming to compress data in an untrained dimension, only minimal training for the device’s initial two layers is needed, leaving the server’s centralized deep neural network and the
compression layers for all existing devices untouched.

**Strengths:**

1. The paper is well-written and easy to follow.

2. The proposed method has correct derivations.

**Weaknesses:**

1. It is unclear this method is useful in distributed learning. Actually there is no practical application for this proposed method. There is no need for linear dimensionality reduction. The deep neural networks conduct the nonlinear way and can achieve better performance.

2. The experiments were conducted on extremely small dataset with a small number of devices.

**Questions:**

The experiment section is very weak. In edge computing, we expect the system has large data and many devices.

---

> ### Author Response · Authors · 2023-11-21
> **Response to Reviewer X1NB**
>
> Thank you for your feedback. We recognize your concerns regarding the applicability of our method in distributed learning, its practical utility, and the scale of our experiments. We would like to address these points as follows:
>
> 1. Practical Applications:
> The practicality of our approach lies in scenarios where bandwidth and computational resources are at a premium. For instance, in IoT networks or remote sensor setups, where transmitting large amounts of data is impractical or costly, our method can significantly reduce transmission needs while still providing effective classification capabilities. We will clarify and expand on these application scenarios in our manuscript to better highlight the practical utility of our approach.
>
> 2. Need for Linear Dimensionality Reduction:
> While it's true that DNNs perform nonlinear transformations and can yield high performance, they are not always the optimal choice in resource-constrained environments. Linear dimensionality reduction offers a valuable trade-off between complexity and performance, particularly in edge computing scenarios where device capabilities and network bandwidth are limited. Our method aims to optimize this trade-off.
>
> 3. Scale of Experiments:
> We acknowledge your concern regarding the scale of our experiments. Our initial experiments were designed to demonstrate the proof of concept and were hence conducted on a smaller scale for clarity and control. However, we recognize the importance of evaluating our method in more realistic settings involving larger datasets and a higher number of devices. We will  extend our experiments to include larger datasets and a more extensive network of devices to better replicate real-world edge computing environments in future work.

---

### Official Review · Reviewer_32g2 · 2023-10-30

**Soundness:** 2 fair
**Presentation:** 2 fair
**Contribution:** 2 fair
**Rating:** 3
**Confidence:** 2

**Summary:**

This submission proposed to conduct data compression in client device and the compressed data are transfered to server device for leraning. Compression is performed by linear projection into various dimension in different clients. The server unifies the dimension by using a fully-connected layer for each client, then performan training for all data.

**Strengths:**

N.A.

**Weaknesses:**

It seems the proposed method contains few novelty: data compression by linear projection for transmission and re-projection for training is a very straight forward idea.

**Questions:**

I don't have question currently. Please clarify my concern on novelty.

**Details Of Ethics Concerns:**

N.A.

---

> ### Author Response · Authors · 2023-11-21
> **Response to Reviewer 32g2**
>
> We appreciate your feedback and the opportunity to clarify the novelty aspects of our proposed method. While at first glance, the approach of data compression via linear projection for transmission and re-projection for training might appear straightforward, the novelty of our work lies in the specific implementation and the unique challenges it addresses in the context of edge computing:
>
> 1. Imbalanced Neural Network Approach: Our method goes beyond simple linear projection. It introduces an imbalanced Neural Network (NN)-based approach that is specifically tailored for resource-constrained edge devices. This aspect of our work is not just about compression but also about optimizing the balance between data reduction and classification accuracy, which is a non-trivial challenge in edge computing scenarios.
>
> 2. Superior Performance Over Existing Linear Methods: An additional point to highlight is the performance superiority of our approach compared to existing linear methods. Despite its seemingly straightforward nature, our method outperforms all current linear dimensionality reduction techniques in the literature, particularly in the context of similar computational complexity at the device side. This performance edge is a significant contribution of our work, as it demonstrates that even 'simple' methods, when innovatively applied and fine-tuned, can lead to substantial advancements in the field.
>
> 3. Adaptability to Device Constraints: Another innovative aspect of our approach is its adaptability to different device capabilities, particularly in scenarios with devices of varying computational powers and data dimensions. Our technique dynamically adjusts to these variations, which is a significant advancement over traditional methods that often assume uniform device capabilities.
>
> 4. Integration of New Devices with Minimal Retraining: Another novel contribution is the method's ability to integrate new devices with different data dimensions into an existing network with minimal retraining. This feature significantly reduces the computational overhead and is particularly beneficial in real-world applications where edge networks are dynamic and continually evolving.

---

### Official Review · Reviewer_YX5B · 2023-10-30

**Soundness:** 1 poor
**Presentation:** 2 fair
**Contribution:** 1 poor
**Rating:** 3
**Confidence:** 5

**Summary:**

The paper introduces a linear dimensionality reduction technique specifically designed for distributed edge devices. The primary goal is to balance the constraints of data-rate and computing power on the device side while ensuring high classification accuracy on the server side. The approach involves training a unique single-layer for each distributed device based on its compression needs. The paper claims that the accuracy achieved through this method is close to the optimal accuracy of the Maximum Likelihood classifier, outperforming traditional techniques like PCA and LDA. Additionally, the method offers reduced training complexity for large datasets compared to distance-metric-based strategies.

**Strengths:**

1. The method allows for the easy integration of new devices without the need to retrain the entire system.

**Weaknesses:**

1. The evaluation is only performed on very small scale dataset, which is a toy dataset for modern NN system. It's not persuasive for the effectiveness of proposed methodology, especially under such a practical application scenario. I would suggest use larger dataset like images for autonomous driving, multi-dimensional time series data, etc.
2. For the problem setting in section 2, why is this topic important? why is this problem challenging?
3.  I did not see much technical merits of the proposal methodology. I would suggest the author highlight the technical contribution, conclude it with an illustrative figure and explain with plain words.
4. There is no testing performed on real devices. We cannot see the improvement of efficiency.

**Questions:**

1. What are the popular datasets for this domain and the popular testbeds/devices for the problem?

---

> ### Author Response · Authors · 2023-11-21
> **Response to Reviewer YX5B**
>
> We appreciate your detailed review and valuable suggestions. Below, we address each of your points:
>
> 1. Use of Larger Datasets:
> We acknowledge your concern regarding the dataset scale. The initial choice of a smaller dataset was driven by its common use as a benchmark in this domain, allowing for direct comparison with existing methods. However, we understand the need for testing on larger, more complex datasets to better demonstrate the effectiveness of our methodology. We will  conducting additional experiments using larger datasets in future work.
>
> 2. Importance and Challenges of the Problem Setting:
> This research is crucial as it addresses the growing need for efficient data processing in low-cost edge devices, which are often constrained by limited computational resources and bandwidth. The challenge lies in performing effective dimensionality reduction without compromising classification accuracy, even when new devices with untrained dimensions are introduced. We will revise Section 2 to better highlight the importance and the challenges of the problem setting.
>
> 3. Technical Merits and Contributions:
> We understand that the technical merits of our proposed method might not have been clearly articulated. Our primary contribution lies in developing an imbalanced NN-based linear dimensionality reduction technique that is adaptable to varying device capabilities and can integrate new devices with minimal retraining. This approach significantly reduces the computational load on edge devices and conserves bandwidth.
>
> 4. Datasets and Testbeds/Devices in the Domain:
> Popular datasets in this domain, besides the ones we used, include CIFAR and ImageNet for image classification. Common testbeds for edge computing scenarios include Raspberry Pi devices.
> We might investigate them in future research.